# Effects on Quality of Life of a Telemonitoring Platform amongst Patients with Cancer (EQUALITE): A Randomized Trial Protocol

**DOI:** 10.3390/mps7020024

**Published:** 2024-03-15

**Authors:** Felipe Martínez, Carla Taramasco, Manuel Espinoza, Johanna Acevedo, Carolina Goic, Bruno Nervi

**Affiliations:** 1Centro para la Prevención y Control del Cáncer (CECAN), Santiago 8331150, Chile; carla.taramasco@unab.cl (C.T.); manuel.espinoza@uc.cl (M.E.); johannaacevedo@udd.cl (J.A.); cgoicb@uc.cl (C.G.); bnervi@uc.cl (B.N.); 2Facultad de Medicina, Escuela de Medicina, Universidad Andrés Bello, Viña del Mar 2531015, Chile; 3Concentra Educación e Investigación Biomédica, Viña del Mar 2552906, Chile; 4Facultad de Ingeniería, Universidad Andrés Bello, Viña del Mar 2531015, Chile; 5Departamento de Salud Pública, Pontificia Universidad Católica de Chile, Santiago 8330023, Chile; 6Unidad de Evaluación de Tecnologías en Salud, Centro de Investigación Clínica, Pontificia Universidad Católica de Chile, Santiago 8330023, Chile; 7Instituto de Ciencias e Innovación en Medicina, Universidad del Desarrollo, Santiago 7550000, Chile; 8Facultad de Medicina, Escuela de Medicina, Pontificia Universidad Católica de Chile, Santiago 8330023, Chile; 9Foro Nacional del Cáncer, Santiago 8340696, Chile; 10Departamento de Hematología y Oncología, Escuela de Medicina, Pontificia Universidad Católica de Chile, Santiago 8330023, Chile

**Keywords:** cancer, oncology, telemonitoring, patient surveillance, internet, smartphone, quality of life, toxicity, adverse events

## Abstract

Cancer, a pervasive global health challenge, necessitates chemotherapy or radiotherapy treatments for many prevalent forms. However, traditional follow-up approaches encounter limitations, exacerbated by the recent COVID-19 pandemic. Consequently, telemonitoring has emerged as a promising solution, although its clinical implementation lacks comprehensive evidence. This report depicts the methodology of a randomized trial which aims to investigate whether leveraging a smartphone app called *Contigo* for disease monitoring enhances self-reported quality of life among patients with various solid cancers compared to standard care. Secondary objectives encompass evaluating the app’s impact on depressive symptoms and assessing adherence to in-person appointments. Randomization will be performed independently using an allocation sequence that will be kept concealed from clinical investigators. Contigo offers two primary functions: monitoring cancer patients’ progress and providing educational content to assist patients in managing common clinical situations related to their disease. The study will assess outcomes such as quality of life changes and depressive symptom development using validated scales, and adherence to in-person appointments. Specific scales include the EuroQol Group’s EQ-5D questionnaire and the Patient Health Questionnaire (PHQ-9). We hypothesize that the use of Contigo will assist and empower patients receiving cancer treatment, which will translate to better quality of life scores and a reduced incidence of depressive symptoms. All analyses will be undertaken with the intention-to-treat principle by a statistician unaware of treatment allocation. This trial is registered in ClinicalTrials under the registration number NCT06086990.

## 1. Introduction

Cancer presents a significant and persistent challenge to global health, ranking among the leading causes of death on a worldwide scale. In 2020, it claimed nearly 10 million lives, as reported by the World Health Organization [1]. The surge in cancer cases can be attributed to factors such as population growth and aging, alongside shifts in the prevalence and distribution of major cancer risk factors, such as obesity, tobacco and alcohol consumption [2]. Significantly, these risk factors are intertwined with socioeconomic development. Predominant forms of cancer encompass breast, lung, colon, rectal, and prostate cancers [3]. Chemotherapy and radiotherapy commonly emerge as treatment modalities for many instances of these cancer types. Roughly 50% of all cancer patients receive radiotherapy as part of their treatment [4], and in 40%, radiotherapy is used as a part of treatments with curative intent. A recent population-based study [5], utilizing data from GLOBOCAN 2018, identified a substantial demand for the use of chemotherapy, which is projected to expand by 53% by the year 2040. The disparity between the available service provision and the growing demand was particularly significant in low-income and middle-income countries.

Presently, the majority of radiotherapy and chemotherapy treatments are administered to patients as outpatients [6,7]. Follow-up care is typically conducted in person, but there is notable variability in this approach across different clinical centers. Meticulous patient surveillance during chemotherapy and radiotherapy is essential for effective cancer treatment. These potent therapies, while targeting cancer cells, can also affect healthy tissues, causing various side effects. Although many of these adverse events can be managed in the outpatient setting, such as nausea, vomiting, or fatigue, some are severe enough to merit admission to a hospital. Examples of the latter potentially life-threatening complications include neutropenic fever, cytopenias, and septic shock [8,9].

Traditional in-person follow-up strategies in oncology have several limitations, some of which were highlighted by the recent COVID-19 pandemic [10,11]. These strategies demand significant healthcare resources, including time, infrastructure, and expenses, posing a strain on healthcare systems [12,13,14]. Geographical barriers can hinder access for patients in remote areas, causing delays in follow-up care. In-person appointments are often inconvenient for patients due to travel, scheduling conflicts, and associated stress, potentially leading to missed appointments. The frequency of follow-ups may be suboptimal, delaying the detection of disease recurrence or relevant side effects attributable to chemotherapy or radiotherapy regimens. Additionally, these visits may not capture a comprehensive view of the patient’s health between appointments, impacting treatment decisions [15]. Long waiting times, reduced privacy, infection risks during pandemics, and inefficient data collection are other drawbacks. To address these limitations, telemedicine and remote monitoring technologies are being increasingly incorporated to enhance accessibility, convenience, and efficiency of follow-up care while maintaining high standards of patient support [16].

In essence, telemonitoring entails the transmission of pertinent clinical information through digital platforms, encompassing internet-connected medical devices like smartphones, health tracking apps, and video conferencing tools. These technologies not only enable the monitoring of diverse clinical conditions but also serve as a channel for continuous communication between patients and healthcare professionals [17,18]. The primary objective is to deliver precise and timely medical attention, thereby enhancing the patient’s quality of life and curbing treatment expenses by reducing the necessity for frequent in-person visits. Telemonitoring platforms are especially beneficial for individuals facing challenges in attending regular in-person medical check-ups or those residing far from healthcare facilities. Moreover, the potential reduction in healthcare visits can lead to a less disruptive daily routine for patients, facilitating better social integration and an improved quality of life [19,20].

Prior experiences with adult cancer patients have demonstrated the feasibility and effectiveness of various telemonitoring approaches in enhancing patient care. Early studies indicate notable advancements in symptom detection, such as pain [21,22], improvements in quality of life [23], and enhanced detection of post-surgical complications [24,25]. While these initial findings are promising, the exploratory and pilot nature of these studies somewhat limits their definitive conclusions. We therefore designed this study to assess the efficacy of a telemonitoring platform using a smartphone application amongst patients with cancer.

## 2. Objective

The primary objective of this study is to ascertain whether, among patients with various forms of solid cancer, the use of a smartphone application designed to facilitate disease monitoring leads to an improvement in the quality of life as reported by the patients themselves compared to standard care. As secondary objectives, we aim to determine the effects of this application on the presence of depressive symptoms and assess the level of adherence to in-person appointments established by the treating team.

## 3. Methodology

To fulfill the aforementioned objectives, a randomized parallel-group clinical trial will be conducted among patients with a recent diagnosis of cancer undergoing curative treatment at the UC-Christus Cancer Center. This center features an outpatient oncology clinic providing specialty care for patients belonging to the public and private systems in Chile. This protocol has been drafted following the Consolidated Standards of Reporting of Randomised Trials (CONSORT) guidelines [26] and also adheres to CONSORT-EHEALTH for web-based and mobile health intervention recommendations [27]. The study flowchart is depicted in Figure 1.

### 3.1. Participants

Eligible participants will be adult patients (≥18 years old) with a recent histologically confirmed diagnosis (within the last 3 months) of bronchogenic, breast, gallbladder, gastric, colorectal, or prostate cancer in any of its forms. These patients should be awaiting the initiation of curative intent treatment for the disease using any modality (radiation therapy, chemotherapy, immunotherapy, etc.) at the UC-Christus Cancer Center during the months of November 2023 to June 2024. It was decided not to include patients with hematological neoplasms at this stage because most chemotherapy regimens provided for these forms of cancer are delivered within hospitals in Chile. Participants will be required to possess a smartphone, regardless of its native operating system (iOs^®^ or Android^®^). All selected patients will be required to sign an informed consent form to participate in this study. Individuals with any form of sensory impairment preventing app usage, cognitive impairment, psychiatric pathology hindering app usage, or those unwilling to participate in the study will be excluded. Patients concurrently participating in another clinical trial addressing healthcare technologies will be excluded as well. 

In brief, the recruitment process will proceed as follows. After confirming the cancer diagnosis by the treating team, a message will be sent via email and/or a phone call to inform the possibility of participating in the study. This contact will not occur within the first 72 h after the visit with the treating medical team. If there is an interest in participating in the study, a formal clinical visit will be scheduled by the study personnel (nursing technician or nurse) on the same day as the routine check-ups established by the treating team. During this visit, inclusion criteria compliance and absence of exclusion criteria will be re-evaluated. The study’s characteristics, procedures, potential benefits and harms will be explained. Some examples of potential benefits include enhanced ease of communication with the treating team, improved record-keeping of information, and the detection of chemotherapy-associated toxicity events, ultimately leading to an improvement in the quality of life resulting from these advancements in patient monitoring. However, there is also the potential to induce increased anxiety or a hypervigilant attitude regarding cancer treatment, which will also be discussed with each potential participant. The healthcare team responsible for providing clinical services to the patients will not directly invite them to the study. The invitation will outline the interventions, potential benefits, participation requirements, and clarify that the healthcare team delivering clinical services to patients will not issue direct invitations to the study. 

### 3.2. Procedures

Once informed consent is obtained, patients will be randomized in a 1:1 ratio using a permuted block method to receive one of the two intervention strategies. This procedure will be conducted by an investigator independent from patient care using a computer algorithm that will be kept concealed from other study investigators.

Patients allocated to the active intervention group will receive a smartphone application called *Contigo* (see Application Design and Development, below). The application will be delivered with assistance from the study team, including the download process, user account generation, and education regarding the usage of its functionalities. The *Contigo* application aims to fulfill two primary functions: monitoring cancer patients and delivering educational content, providing tools for patients to navigate common clinical situations associated with the diagnosis and treatment of their disease. Examples of such situations include symptoms associated with chemotherapy, health insurance coverage, and implementing a cancer patient’s treatment. The first objective will be achieved through the deployment of modules and sub-modules, where the user will input and record their perceptions and experiences via questionnaires associated with their oncological process. The second objective will be implemented through the delivery of health educational content for cancer patients. This educational content was designed by a team of professionals, including medical oncologists, nurses, and oncology patients from the project’s associated entities, through group sessions (focus group). Apart from this source for design, an analysis of current scientific evidence and official reports or regulations from the Ministry of Health related to educational material for cancer patients will be conducted to complement the elements developed in the group sessions. The topics covered in these educational contents will be specific to each patient’s type of cancer and will include aspects of the healthcare process, administrative aspects, health coverage, self-awareness, and self-care practices.

Participants assigned to the control group will receive standard educational care regarding their illness and in-person check-ups based on the study’s outcome schedule and the determination of their attending physician. However, once the study’s follow-up period is completed, participants assigned to the control group will be offered access to this application. 

### 3.3. Contigo: Application Design and Development

*Contigo* is an application developed under a project funded by public resources from the Agencia Nacional de Investigación y Desarrollo (ANID, FONDAP ID 152220002), led by PhD Carla Taramasco on behalf of Universidad Andrés Bello, who holds the intellectual property rights. The concept behind *Contigo* emerged from a research proposal funded by Pontificia Universidad Católica de Chile, titled “Identification and Measurement of Needs of Oncological Patients and Healthcare Professionals for the Development of New Technological Support Platforms for Patients”. The project’s main objective was to identify information needs for both patients and healthcare professionals through focus groups and interviews.

The obtained results underwent a content analysis process, leading to the proposal of a mobile application for patients addressing specific information needs. *Contigo* was developed as a Progressive Web App (PWA) using web technologies, providing a user experience similar to a native application on any smartphone, regardless of its operating system. A PWA offers features like offline functionality, push notifications, and access to hardware on the device. Some of the key advantages of PWAs are their ability to run on any device with a web browser and not requiring installation from an app store. This also makes them lighter and faster than native applications since they run in the web browser and do not need to be fully downloaded onto the device. The SCRUM agile software development methodology was employed, featuring a client–server architecture and modular design for scalability and ease of incorporating new functionalities.

*Contigo* comprises six modules: El Viaje (Spanish for The Journey, care process information), Mis Resultados (My Results, self-perception of health reporting), Mis Experiencias (My Experiences, reporting experiences during healthcare and cancer treatment), Asistencia (Assistance, frequently asked questions), Comunidad (Community, complementary non-clinical information), and Programación (Scheduling, medical appointment and procedure requests). Users have continuous access to these modules, and questionnaires are displayed according to the clinical trial schedule. The source code will not be published, but a website (https://infanticontigo.cl/, accessed on 3 December 2023) will provide information about the application and a functional demo version for potential users. Patients participating in the clinical trial will receive the application at no cost, with usernames and passwords provided during training by the monitoring nurse. The research team will offer technical support, while clinical assistance will be provided by healthcare professionals. A depiction of the user interface of *Contigo* is depicted in Figure 2.

*Contigo* is founded upon a robust security infrastructure designed to safeguard the integrity and confidentiality of data. HTTPS is employed to ensure the secure transmission of information between the client and the server. Passwords are stored in encrypted form, and hash techniques are applied to protect sensitive database data. Environment variables and system keys are generated during the execution of the deployment pipeline, preventing their inclusion in configuration files or source code, and ensuring they remain unknown to developers. Through these measures, a secure environment is maintained to preserve the privacy of users and the integrity of stored information.

### 3.4. Variables

For each patient enrolled in the study, a baseline clinical profile will be developed, including demographic, household characterization, clinical, and neoplasia-related information. Briefly, this profile will consist of demographic data, encompassing information about gender, age, highest level of education attained, marital status (single, married, cohabiting), healthcare system affiliation (Fondo Nacional de Salud, FONASA, public healthcare; Institución de Salud Previsional, ISAPRE, private insurance; or other); household characterization data including socioeconomic level, overcrowding, availability of basic services, migratory status, and support networks; and clinical information. The latter section will comprise data on relevant medical comorbidities (diabetes mellitus, heart failure, chronic obstructive pulmonary disease, bronchial asthma, chronic liver disease, fibromyalgia, coronary artery disease, and chronic kidney disease), concurrent psychiatric conditions (depression, anxiety disorders, personality disorders), habits (alcohol consumption, substance use, or smoking), and neoplasia-related information, including the specific cancer type, date of diagnosis, stage at diagnosis based on TNM (tumor, nodes, and metastasis) classification, and therapy received.

The participants’ health-related quality of life will also be assessed using the EuroQol Group’s EQ-5D questionnaire. This tool evaluates an individual’s perception of their health status based on well-being in five dimensions: mobility, self-care, usual activities, pain/discomfort, and anxiety/depression. Each of these dimensions is subdivided into five response levels for each question, and scores can be assigned to synthesize an individual’s quality of life. In addition to the questions for each dimension, the EQ-5D questionnaire includes a synthesis measurement represented by a visual analog scale regarding the individual’s self-perceived current health state. The scale is a 20 cm line graduated in units where the top end represents 100 points denoting the “best imaginable health state”, and the bottom end represents 0 points indicating the “worst imaginable health state”. The respondent evaluates their current health state by drawing a line from the box marked “your own health state today” to the point that best represents their health on this visual analog scale. This questionnaire was chosen for its validation for use among oncology patients [28], as well as its favorable psychometric properties and ease of use, enabling self-administration, validation in multiple languages including Spanish, and application among Chilean patients [29,30]. All these baseline measurements will be performed once during the initial study visit by the research team, taking the form of an initial clinical interview.

### 3.5. Outcomes

The primary outcome of this study is the change in quality of life quantified using the EQ-5D questionnaire measured at 30, 60, and 90 days after randomization. The five-level version of the questionnaire will be used (EQ-5D-5L). Additionally, the measurement of quality of life specific to the 3 domains established by the previous questionnaire has been established as co-primary outcomes. The questionnaire will be electronically delivered through the mobile application or via emails for participants to self-administer, considering its psychometric properties, when reaching the aforementioned follow-up milestones of the study. If electronic delivery of the questionnaire is not possible, it can be completed during in-person visits of the study participants.

As a secondary outcome, the difference in the development of depressive symptoms during the follow-up has been established. These symptoms will be evaluated based on the Patient Health Questionnaire 9 (PHQ-9) scale [31]. This diagnostic tool consists of 9 questions assessing the level of depression in an individual. The first 8 questions accumulate a score ranging from 0 to 3 points, where higher scores suggest a higher probability of depression. The ninth question is used to determine the impact of the patient’s symptoms on their personal life. Scores equal to or greater than 10 points demonstrate good diagnostic performance, with a sensitivity of 88% (95% Confidence Interval, CI 95% 83–92%) and specificity of 85% (CI 95% 82–88%) based on a recent systematic review [32]. This scale was chosen not only for its clinical performance but also for its extensive use in various clinical settings [31,33,34] and spectrums of chronic comorbidities [35,36]. It is important to note that the questionnaire is validated for use in the Spanish language and in the Chilean population [37]. Additionally, this tool is designed to be applied without the need for a healthcare provider’s intervention [38,39], thus it can be virtually implemented through the same application within the same time frames as the EQ-5D tool. In case of a positive screening for depression using the aforementioned scale, the treating team will be notified of the result for confirmation and treatment inclusion in participant management. Also, adherence to medical appointments established by the treating team will be evaluated as a secondary outcome, defined as the proportion of in-person attendance at medical and nursing appointments during the study period.

### 3.6. Statistical Analysis

#### 3.6.1. Sample Size

Current literature on the potential effect of telemonitoring platforms on outcomes such as quality of life is scarce. However, based on previous reports [40,41], it has been estimated that a total of 80 participants (40 per arm) will be necessary to achieve 80% statistical power at standard significance levels (5% alpha two-tailed), assuming an intergroup difference in EQ-5D questionnaire means of 0.1 point, a symmetric standard deviation between groups of 0.35 points, a strength of linear correlation between measurements of 0.7 (r = 0.7), and 3 instances of measurement for both groups.

#### 3.6.2. Analysis Strategy

For the statistical analysis, means, medians, standard deviations, interquartile ranges, ranges, and absolute and relative frequencies will be used for descriptive analysis. Inferential analysis will employ Student’s *t*-test or the Mann–Whitney U test for comparing means based on data distribution and variances, and Fisher’s Exact Test for evaluating categorical variables. The primary outcome will be evaluated in relation to the change in EQ-5D score means associated with their corresponding standard deviations through a repeated measures analysis of variance (ANOVA). Confidence intervals at 95% will also be constructed from the standard deviations of the mean changes to estimate the magnitude of the intervention effect among these patients. Additionally, the potential association between the implemented intervention and the development of categorical outcomes will be quantified using the relative risk statistic associated with their corresponding 95% confidence intervals. Subgroup analysis has not been considered in this study.

If an imbalance in participant characteristics is detected, confounding will be controlled using multivariable linear regression analyses. Model assumptions will be assessed graphically and by an evaluation of residuals. Heteroskedasticity will be addressed using the Cook–Weisberg method. All analyses will be conducted by a statistician unaware of the participants’ treatment assignment, following the intention-to-treat principle, and using Stata v.16.0^®^ software (StataCorp LP, StataCorp LLC., College Station, TX, USA, 1996–2020).

## 4. Research Ethics

This study has undergone ethical evaluation by the Research Bioethics Committee of the Pontificia Universidad Católica de Chile, accredited by the Chilean Ministry of Health. Its approval number is 012793. This protocol has been registered in a ClinicalTrials.gov (accessed on 3 December 2023), a publicly accessible repository in accordance with the international standards set by the International Committee of Medical Journal Editors regarding the conduct of clinical trials. Its registration number is NCT06086990.

## 5. Conclusions

This study protocol outlines an innovative approach to address challenges in cancer care, specifically focusing on chemotherapy for various solid carcinomas. The *Contigo* app introduces a cutting-edge telemonitoring platform that could significantly improve the way in which follow-up is conducted amongst patients with cancer. This platform is designed to monitor chemotherapy toxicity symptoms and offer educational content to cancer patients.

The necessity for this tech-driven solution became apparent due to the limitations exposed by the COVID-19 pandemic of traditional follow-up approaches, underscoring the importance of remote monitoring in healthcare. *Contigo* strives to adopt a patient-centric approach by prioritizing patient experiences during chemotherapy. This is achieved through validated questionnaires and a multidisciplinary perspective. Among its potential clinical benefits is the generation of a novel remote monitoring alternative, aiming to alleviate the workload of oncology services by enabling the remote surveillance of patients. Additionally, this proposal has the potential to enhance the clinical monitoring process, increasing the detection rate of complications during treatment and improving assessment frequency by adapting to the patient’s needs. This platform also streamlines the utilization of validated tools for detecting symptoms and complications in oncology patients, potentially contributing to the enhancement of control quality and facilitating research development in the field.

The application comprises modules that encompass vital aspects of cancer care, allowing patients to report their health status and experiences, access educational materials, and connect with community resources. Notably, this study aims not only to improve the patient experience, but also to assess its impact on healthcare providers. This includes evaluating clinician satisfaction and its potential influence on patient care.

While the protocol outlines robust plans for data collection and statistical analysis, it acknowledges certain limitations such as sample size and specific cancer types being targeted. Its randomized design will mitigate the possibility of selection and confusion biases, and the use of validated questionnaires reduces the possibility of systematic errors due to information bias. Nevertheless, it lays the groundwork for a promising study that could significantly advance cancer care by leveraging technology to bridge gaps in patient monitoring and support.

## Figures and Tables

**Figure 1 mps-07-00024-f001:**
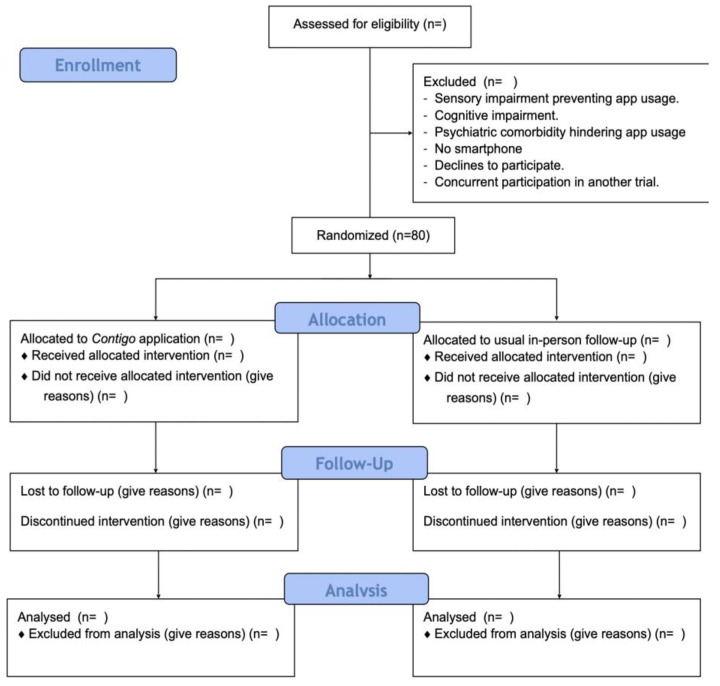
CONSORT flow diagram for the EQUALITE study.

**Figure 2 mps-07-00024-f002:**
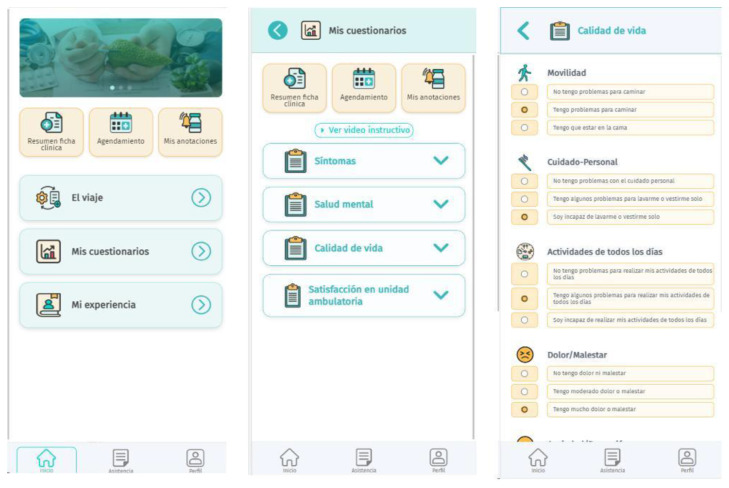
Contigo application user interface.

## Data Availability

No new data are currently available for data sharing. However, upon completion of the trial, data will be made available to researchers upon reasonable request by contacting the authors.

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
