# Peer review of "Effects on Quality of Life of a Telemonitoring Platform amongst Patients with Cancer (EQUALITE): A Randomized Trial Protocol"

_mps, 2024, doi:10.3390/mps7020024_

Round 1

Reviewer 1 Report

Comments and Suggestions for Authors

The manuscript "Effects in Quality of Life of a Telemonitoring Platform Amongst Patients With Cancer (EQUALITE): A Randomized Trial Protocol" presents novel information. However, it is necessary to make some adjustments to give the information in a better way and to make it more straightforward for the readers.

- The title is clear. However, it could include a direct reference to telemonitoring as a first approach. I think the abstract should be revised because it is necessary to mention the results achieved and/or expected briefly.

- Introduction: The introduction has a direct context to what the problem is. However, a more detailed explanation could be given as to why in-person monitoring is limited and how telemonitoring addresses these challenges. Especially since the pandemic is "over," while they mention the issue of mobility and other physical problems, they then end up contradicting it by indicating that they removed people from the study who did not have a smartphone or the ability to use one, which is the norm in the most remote, impoverished or elderly populations who suffer the most from these types of pathologies; not to mention people with mobility restrictions or psychological issues. I would suggest you make a more detailed justification of why these population sectors are being eliminated without providing any future solution.

- Methodology: It would be helpful to provide information on how specifically the results will be measured and how possible biases will be handled; that would give it more impact, as well as how it could subsequently include people who refused.

- Results: This section should be restructured as no information is provided on the actual results of the study. I suggest mentioning something hypothetical that could happen since the authors only mention the method and how they expect to apply it. This could apply to discussion as well.

- Conclusion: I suggest that the conclusion should be more emphatic and highlight the relevance of the actual possible results of the study and how they could influence cancer care.

General comments:

It is essential to include actual results in the article and to ensure that the authors are supported by sound statistical analysis. It would also be important to provide more detail on the clinical significance and practical implications of the results.

Comments on the Quality of English Language

 Minor editing of English language required

Author Response

Dear Reviewer,

Thank you very much for your work and for providing feedback on our research protocol. We believe that your insights have been valuable and have allowed us to enhance the quality of our proposal.

We have reviewed your suggestions and implemented several changes in our text. However, there are some discrepancies in relation to the suggestions, primarily stemming from the fact that our study is a research protocol. As such, we are not yet in a position to share or discuss results. However, this will be done as soon as we have completed the recruitment and follow-up of participants.

Please find a detailed response to your comments in the attachment below. 

We sincerely appreciate your contributions and hope that our responses help mitigate your concerns. 

Kind regards,

Reviewer 2 Report

Comments and Suggestions for Authors

I have carefully reviewed your work and find it to be potentially valuable in addressing current healthcare needs in your social context. However, there are some aspects that I would like to discuss with you for potential improvement:

Introduction: While the introduction appears to be centered on the necessity of new technologies in cancer patient monitoring, it lacks sufficient bibliographic references to support its claims, as observed from my perspective.

The statement regarding the rapidly escalating incidence and mortality rates of cancer may benefit from clarification, as these values have stabilized over the last decade. Additionally, the reference to chemotherapy and radiotherapy as emerging modalities might need further substantiation.

The phrase "They demand..." at line 59 requires clarification regarding the specific entities referred to. As mentioned earlier, such assertions should be supported by corresponding literature.

The objectives exclusively focusing on solid tumours raise the question of why not include both solid and soft tumours. A brief explanation for this choice would enhance clarity.

Methodology: The decision to include participants aged 19 and above could benefit from justification. Why not consider individuals from 18 years and older?

The statement that "the potential benefits will be explained" in line 131 is made but not fulfilled in the outlined protocol. Additionally, the potential negative effects are not addressed.

In relation to the proposed evaluation of questionnaires (line 150), please provide a comprehensive list of all variables and questionnaires to be used. The mention of EQ-5D and EORT QLQ-C30 raises questions, especially as the latter is not explained or confirmed for evaluation.

The decision to employ a generic questionnaire such as EQ-5D requires a more robust justification. Why choose EQ-5D over alternatives like SF-36? Please specify the version of EQ-5D selected.

Consideration of Additional Variables in Analysis: I recommend considering the inclusion of other variables, alternative analyses, and the incorporation of confounding or control variables in your study.

Ethical Considerations: It would be beneficial for the authors to include an ethical considerations section, detailing how data collected in the application will be treated, ensuring compliance with relevant data protection standards and current legislation.

Comments on the Quality of English Language

I would recommend considering a review of the English language in specific sections of your work to enhance clarity and coherence.

Author Response

Dear Reviewer,

Thank you very much for your work and for providing feedback on our research protocol. We believe that your insights have been valuable and have allowed us to enhance the quality of our proposal.

We have reviewed your suggestions and implemented several changes in our text. Please find our responses in the attached file, below. We hope that this new version adequately addresses your concerns regarding our study. 

We sincerely appreciate your contributions.

Reviewer 3 Report

Comments and Suggestions for Authors

I would like to commend the authors for their efforts in conducting this study and their contribution to the field of cancer care. Telemonitoring platforms have the potential to significantly impact the quality of life of patients with cancer, and this research is timely and important. I believe that, with some minor revisions to improve clarity, the manuscript could be a valuable contribution to the field.

My suggestions for improving clarity:

Line 20: Consider adding the word "treat" after "radiotherapy" to make it clearer: "Cancer, a pervasive global health challenge, necessitates chemotherapy or radiotherapy treatments for many prevalent forms."

Line 29: You could clarify the sentence a bit: "Contigo offers two primary functions: monitoring cancer patients' progress and providing educational content to assist patients in managing common clinical situations related to their disease."

Line 41-42: Consider breaking down the information in sentence 3 for better readability: "In 2020, cancer claimed nearly 10 million lives worldwide, as reported by the World Health Organization."

Line 42-43: You could specify some of the major cancer risk factors to provide more context: "The surge in cancer cases can be attributed to factors such as population growth, an aging population, and shifts in the prevalence and distribution of major cancer risk factors."

Line 50-51: You might consider rephrasing for improved flow: "Follow-up care is typically conducted in person, but there is notable variability in this approach across different clinical centers."

Line 64-65: Consider rephrasing to make it more concise: "The infrequency of follow-up visits may delay the detection of disease recurrence or relevant side effects from chemotherapy or radiotherapy regimens."

Comments on the Quality of English Language

I have included comments on the English Language in the Comments section.

Author Response

Dear Reviewer,

Thank you very much for your work and for providing feedback on our research protocol. 

We have reviewed your suggestions and implemented several changes in our text. Please find our responses in the attached file, below. We hope that this new version adequately addresses your concerns regarding our study. 

We sincerely appreciate your contributions.

Reviewer 4 Report

Comments and Suggestions for Authors

Contigo app will help cancer patients in many aspects of quality of life. I don't know if it includes spiritual well-being and social aspects.

Author Response

Dear Reviewer,

Thank you for taking your time to read our research protocol. Please find enclosed a response to your comment below. 

Kind regards,

EQUALITE authors.

Round 2

Reviewer 2 Report

Comments and Suggestions for Authors

Line 259 to 264: are they going to be reported data or measured by questionnaire?

In Figure 2, the EORTC QLQ C-30 questionnaire is shown. Please modify the image accordingly and include it in the same language as the rest of the manuscript.

Comments on the Quality of English Language

-

Author Response

We appreciate the reviewer for these additional comments. We address the raised observations in the editorial process in the text, below:

Line 259 to 264: are they going to be reported data or measured by questionnaire?

The clinical features information pertaining to the participants will be recorded based on the initial clinical interview report of the study, as reported data.

In Figure 2, the EORTC QLQ C-30 questionnaire is shown. Please modify the image accordingly and include it in the same language as the rest of the manuscript.

We appreciate the reviewer's thoroughness in the review process. Indeed, the figure displays an old version of the application where the QLQ-C30 questionnaire had been considered. The latter is no longer part of the methodology of the study. We have replaced the figure to prevent any confusion among readers.

We hope these responses help reconsider our manuscript for publication. 

Kind regards,

Felipe Martinez on behalf of the authors